# Effect of Group Mixing and Available Space on Performance, Feeding Behavior, and Fecal Microbiota Composition during the Growth Period of Pigs

**DOI:** 10.3390/ani14182704

**Published:** 2024-09-18

**Authors:** Adrià Clavell-Sansalvador, Raquel Río-López, Olga González-Rodríguez, L. Jesús García-Gil, Xavier Xifró, Gustavo Zigovski, Juan Ochoteco-Asensio, Maria Ballester, Antoni Dalmau, Yuliaxis Ramayo-Caldas

**Affiliations:** 1Animal Breeding and Genetics Program, Institute of Agrifood Research and Technology (IRTA), Torre Marimon, 08140 Caldes de Montbui, Barcelona, Spain; olga.gonzalez@irta.cat (O.G.-R.); maria.ballester@irta.cat (M.B.); 2Animal Welfare Subprogram, Institute of Agrifood Research and Technology (IRTA), 17121 Monells, Girona, Spain; raquel.rio@irta.cat (R.R.-L.); juan.ochoteco@irta.cat (J.O.-A.); antoni.dalmau@irta.cat (A.D.); 3Digestive Diseases and Microbiota Group, Biomedical Research Institute of Girona (IDIBGI), 17190 Girona, Girona, Spain; jesus.garcia@udg.edu; 4New Therapeutic Targets Lab Research Group, Medical Sciences Departament, Faculty of Medicine, Universitat de Girona, 17071 Girona, Girona, Spain; xavier.xifro@udg.edu; 5School of Medicine and Life Sciences, Graduate Program in Animal Science, Pontifícia Universidade Católica do Paraná (PUCPR), Curitiba 80215-901, Paraná, Brazil; gustavozipaula@gmail.com

**Keywords:** stress, welfare, porcine, feed efficiency, microbiota, biomarkers, gut-brain axis

## Abstract

**Simple Summary:**

Prolonged stress negatively affects pig health, welfare, and productivity. Herein, we used a porcine model of stress during the growing period, divided into stressed and control groups. Stressed pigs experienced reduced space and were mixed twice, leading to decreased body weight and feed efficiency. Differences in feeding behavior were also observed; stressed pigs visited feeders less frequently and spent more time per meal. The microbiota of stressed pigs showed an increase in opportunistic bacteria, while control pigs had a higher abundance of beneficial butyrate- and propionate-producing bacteria. This study highlights the potential of using specific fecal microorganisms as non-invasive biomarkers to assess stress and well-being in pigs, with implications for improving both animal welfare and research applied to the human gut-brain axis.

**Abstract:**

Stress significantly affects the health, welfare, and productivity of farm animals. We performed a longitudinal study to evaluate stress’s effects on pig performance, feeding behavior, and fecal microbiota composition. This study involved 64 Duroc pigs during the fattening period, divided into two experimental groups: a stress group (*n* = 32) and a control group (*n* = 32). Stressed groups had less space and were mixed twice during the experiment. We monitored body weight, feed efficiency, feeding behavior, and fecal microbiota composition. Compared to the control group, the stressed pigs exhibited reduced body weight, feed efficiency, fewer feeder visits, and longer meal durations. In the fecal microbiota, resilience was observed, with greater differences between groups when sampling was closer to the stressful stimulus. Stressed pigs showed an increase in opportunistic bacteria, such as *Streptococcus*, *Treponema* and members of the *Erysipelotrichaceae* family, while control pigs had more butyrate- and propionate-producing genera like *Anaerobutyricum*, *Coprococcus* and *HUN007*. Our findings confirm that prolonged stress negatively impacts porcine welfare, behavior, and performance, and alters their gut microbiota. Specific microorganisms identified could serve as non-invasive biomarkers for stress, potentially informing both animal welfare and similar gut-brain axis mechanisms relevant to human research.

## 1. Introduction

Stress is a body’s non-specific response to challenges that could threaten homeostasis, health, and well-being [1]. The stress response can be the result of several types of stressors, including environmental factors and social challenges. Social factors such as isolation, crowding, and social instability have been shown to induce physiological and behavioral stress responses in livestock [2]. Pigs, like humans, are social animals adapted to living in complex social networks, and as happens with humans, some of their stressors originate from their conspecifics. Pigs form hierarchies at 1–2 weeks of age when they compete to establish a teat order [3], and whenever these animals are regrouped with unfamiliar ones, they tend to exhibit aggressive behavior, allowing them to develop new hierarchies [4]. Therefore, post-mixing aggression could be considered an acute factor of stress that compromises welfare and profitability [5]. Another source of social conflict is related to competition for resources such as food, water, or resting areas [6]. Space is essential in intensive pig production, as animals tend to stay in crowded environments. According to the current EU regulation, a pig of 90 kg should have 0.65 m^2^ [7] but the literature suggests that pigs with a body weight between 66 and 124 kg should have 0.91 m^2^, resulting in a higher average daily gain (ADG) [8]. The greater the deviation from this ideal space allowance, the higher the competition among pigs becomes. This can be a significant source of social stress, particularly as the animals grow larger [9].

Maintaining good welfare standards is crucial to ensure optimal pig production. High stress levels and poor welfare can negatively affect many factors related to pig production including growth rates, feed efficiency, immune function, reproductive performance, and health [10]. For example, glucocorticoids, which increase due to stress, stimulate the hypothalamus to secrete somatostatin, inhibiting the secretion of growth hormone (GH) from the anterior pituitary, affecting animal growth [11]. The reason is that the energy demand increases when the animal copes with a stress factor, and consequently, the available energy for functions like growth is reduced. This decrease in growth could be reflected in reduced productivity and the development of behaviors like tail-biting or excessive fighting [12]. Food intake, daily gain, and body weight affect pig performance [13]. Moreover, stressful situations can reduce the normal function of the immune system, even suppressing the response after vaccination [14].

In addition, chronic stress often affects the gastrointestinal tract, disturbing the microbiome-gut-brain axis and impairing gut barrier integrity, microbiome function, and metabolism [15,16]. Gut bacteria, which regulate the immune system, can increase the risk of infection or autoimmune disease in dysbiosis [17]. Moreover, commensal bacteria may develop an opportunistic phenotype under stressful conditions through pathoadaptive mutations, which allow these microorganisms to colonize and survive more efficiently. Pathoadaptive mutations mainly increase the fitness of their microenvironment by improving nutrient use, aiding immune evasion, or promoting biofilm formation for effective colonization [18]. Furthermore, depression may increase the dominance of proinflammatory species over health-promoting species [19], and stress and depression can improve gut barrier permeability, resulting in a “leaky gut” that allows bacteria to get into circulation, producing a more significant inflammatory response [20]. Other sources of stress, such as birth, weaning, heat stress, and transport stress, also threaten the intestinal health of pigs, leading to disruptions in the gut microbiota [21,22,23].

Human biomedical research benefits greatly from the use of animal models. Pigs’ anatomy, immunology, genome, and physiology are close to humans’ [24]. For example, the pig brain shares similarities with the human brain, being gyrencephalic, and exhibits comparable features of structure, vascularization, anatomy, growth, and development. The gut microbiome of pigs and humans exhibits around a 96% similarity in functional pathways [25]. Since the microbiota-gut-brain axis plays a significant role in neuropsychiatric disorders [26], proposing an animal model that reflects these human-like characteristics is crucial. Previous research has shown that porcine microbiota-gut-brain interactions are comparable to those in humans, indicating that pigs can be effectively utilized to study the gut microbiota changes after neurological events such as stress [27]. However, we have identified limitations in current research by examining how stress affects porcine performance, well-being, and gut microbiota. For instance, as in murine models, gender bias exists, with previous studies primarily focusing on males [28], neglecting potential differences in response to stress across genders. Additionally, a lack of longitudinal studies was noticed, thus failing to capture the dynamic nature of gut microbiota resilience, as the microbial composition may shift significantly over time in response to stress.

In the present study, a social stress model in pigs was used based on two handling procedures: half of the pigs tested were mixed twice during the growing period, and a reduced space allowance was maintained to induce more competition for resources. In comparison, the other half had more space available and were not mixed. This stress model was already tested with success in a previous study [29], where it was found that growing rates differ dramatically between stressed and not-stressed pigs. Therefore, although behavioral and phenotypic differences are also expected in the present study, its main objective is to check the fecal microbiota impacts of this long-term stress model in pigs to identify potential non-invasive fecal microbial biomarkers indicative of animal welfare. These biomarkers could offer insights into the dynamics of the gut microbial ecosystem under prolonged stress situations, to better understand the gut-brain axis, potentially leading to applications in both livestock production and even human health.

## 2. Materials and Methods

### 2.1. Experimental Design and Sample Collection

Animal care and experimental procedures were carried out following national and institutional guidelines for Good Experimental Practices and were approved by the IRTA Ethical Committee (code: 10329). This study raised 64 healthy Duroc pigs of the same genetic line (*Selección Batallé*) at the IRTA experimental farm in Monells (Girona, Spain). The experiment lasted 131 days, from June (animals were 60 days old) to October. The average weight of the animals at their arrival was 18 kg, being 18.8 ± 1.78 kg for castrated males and 18.6 ± 1.64 kg for females, and they were maintained until 140 kg. The 64 pigs were housed in the same building, in a total of 8 pens (4 control and 4 stress) containing 8 pigs each (4 castrated males and 4 females). The control pens had a total space of 12 m^2^ (1.5 m^2^ per pig) and the stress pens a total space of 8 m^2^ (1 m^2^ per pig). In both cases they had the same type of electronic feeding station, a slatted floor, and a drinker. In addition, all pens had two pieces of wood as enrichment material. Pigs were vaccinated against Aujeszky disease at 12 and 16 weeks old. In the four pens considered as stress treatment, animals were mixed twice during the growing period (Figure 1). The first mixing of animals for the stress treatment, which consisted of moving just the females among pens, was performed 61 days after the start of the study, and the second one, which consisted of moving just the castrated males among pens, was performed 22 days later. In both cases, for castrated males and females, three animals were changed to the other three pens (one per pen), and the fourth, the smallest female and the smallest castrated male, remained in the original pen. In this way, mixed pigs did not meet again with their original pen mates (Figure 1). All animals had ad libitum access to a commercial cereal-based diet consisting of a starting, growing and finishing formula (Appendix A). 

### 2.2. Performance and Feeding Behaviour Traits

Body weight (BW in kg) was recorded using a weighing scale. Average daily feed intake (ADFI in kg) was recorded using IVOG^®^ feeding stations (Insentec, Markenesse, The Netherlands). This system records feed intake per meal, from which individual daily feed intake was computed as the sum of all meals in a day. To achieve this, the animals were identified using ear electronic tags (e-tags), which provide daily records from each pig, also documenting the number of visits to the feeder and the duration of feeding events (meal duration in seconds). Additionally, each animal underwent individual monthly weigh-ins, totaling five times over the experimental period. These comprehensive datasets were employed to calculate the ADG and feed conversion ratio (FCR).

### 2.3. Fecal Sampling

Fecal samples were collected three times throughout the experiment between 7:00 and 8:00 a.m. Multiple samplings were planned to gain a deeper understanding of the dynamics of the gut microbial ecosystem throughout the pig-fattening period and to assess the effects of short-term stress episodes. The first sampling was performed on 22 July (07_2022), 52 days from the beginning of the experiment, and five days before the first mix. The second sampling was performed on 2 September (09_2022), 17 days after the second mixing event. The final fecal collection was performed on 5 October (10_2022), 50 days after the second mix (Figure 1). On each sampling day, fecal samples were collected in sterile tubes and stored at −80 °C until DNA extraction.

### 2.4. Microbial DNA Extraction, Sequencing and Bioinformatic Analysis

Microbial DNA extraction was conducted systematically one week after sample collection using 250 mg of fecal content with the DNeasy PowerSoil Pro kit (Qiagen, Hilden, Germany) following the manufacturer’s recommendations. DNA concentration and purity were checked with a nanoDrop One spectrophotometer. Extracted DNA of 60 out of the 64 samples was sent to the Centro de Regulación Genómica (Barcelona, Spain) for paired-end (2 × 300 bp) sequencing on a single run of an Illumina MiSeq (Illumina, San Diego, CA, USA). The V3-V4 fragment of the 16S rRNA gene was amplified using the primers FW (5′-CCTACGGGNGGCWGCAG-3′) and RV (5′-GACTACHVGGGTATCTAATCC-3′). Sequences were analyzed with QIIME2 [30]; barcode sequences, primers, and low-quality reads (Phred score < 30) were removed. The quality control process also trimmed sequences based on the expected amplicon length and removed chimeras. Afterwards, the sequences were assembled into amplicon sequence variants (ASVs) at 99% identity. After removing singletons, only those ASVs representing more than 0.0001% of the total abundance were retained. ASVs were classified to the lowest possible taxonomic level from Phylum to Genus based on a primer-specific trained version of the GreenGenes2 Database (released October 2022) [31].

### 2.5. Statistical Analysis and Microbial-Biomarker Identification

Samples were rarefied at 31,731 reads to correct for sequencing depth. The diversity metrics were estimated with the microeco R package [32]. The α-diversity was evaluated based on the Shannon index [33], and the β-diversity was assessed using the Whittaker index [34]. Permutational multivariate analyses of variance (PERMANOVA) were performed to assess the effect of sex [35]. To identify microbial biomarkers associated with state transitions (e.g., control vs. stress conditions), we utilized the NetMoss2 algorithm, which applies a network-based differential abundance approach [36]. This methodology focuses on shifts in microbial network modules rather than comparing the abundance of individual bacteria, allowing the integration of large-scale microbial datasets. The procedure was implemented step-by-step as follows:

Network Inference: The raw genus abundance matrix for each condition was used to infer a co-occurrence network using the SparCC algorithm [37]. This approach is ideal for handling sparse compositional data, ensuring robust network construction. The control and stress networks (six networks in total) were subsequently integrated into two comprehensive networks (one for control and one for stress), and batch effects were corrected using a univariate weighting methodology [36]. Network graphical representation was created with CytoScape [38], and the topological parameters of the network and ‘node’ centralities values were calculated using the CentiScaPe plugin [39]. 

Module Detection: Modules within these networks were identified using the Weighted Gene Co-Expression Network Analysis. This step allowed the detection of microbial taxa that cooperatively interact within the same module while maintaining competitive interactions between different modules. The connection strength of node *i* defined as the sum of the connections between this node and all other nodes in the network, as:(1)$$k_{i}=\sum_{j=1}^{n}a_{ij}$$

Herein, *a_ij_* represents the correlation coefficient between node *i* and node *j*. While the importance of nodes in the module structure of the network was estimated as:(2)$$k_{ij}=\sum_{j=1}^{n_{1}}a_{ij}-\sum_{j=n_{1}+1}^{n}a_{ij}$$
where *n* represents the number of all nodes in the network and *n*_1_ represents the number of nodes inside a specific module.

Driving Force Calculation: The contribution of each node (bacterial genus) to the transition between the control and stress networks was calculated for each genus using the following model:(3)$$NMSS{\left(i\right)}_{A\to B}=\sum_{j}^{NeighborsA}\Delta D_{ij}-\sum_{l}^{NeighborsB}\Delta D_{ij}$$
where *A* and *B* represent the control and stress networks, respectively. *D* denotes the differential module distance matrix, Neighbors *A* includes all neighboring modules in the control network, and Neighbors *B* represents all neighboring modules in the stress network. The intersection modules represent the stable elements during the transition from health to disease, where the transited modules resulted in alteration of the network structure. In this way, the model quantifies the shift in module structure, highlighting microbial taxa that drive network changes between the experimental conditions. The whole pipeline was performed independently within each sampling time and later conducted longitudinally by integrating the three sampling points (following the NetMoss2 multiple files mode). The FDR method was employed to correct for multiple testing (FDR < 0.05). Finally, the classification performance of the microbial biomarkers was evaluated by using 10-fold cross-validation to implement the ‘netROC’ function within NetMoss2.

## 3. Results

### 3.1. Effect of Stress on Animal Performance and Feeding Behaviour

Animals from the control group were, on average, 7.09% heavier (136 kg; *p* < 0.0001) than those from the stress treatment (127 kg; Figure 2, Table 1). In addition, the control group had a 6.32% extra ADG (1.01; *p* = 0.0001) than the stress group (0.95; Table 1). Differences in feeding behaviour were also detected, with pigs in the control group showing a higher frequency of visits to the feeder (719; *p* = 0.022) than the stress group (589; Table 1). Additionally, pigs from the control group spent less time feeding per day (74.6 min; *p* = 0.028) than their counterparts in the stress group (75.8 min; Table 1).

### 3.2. Impact of Stress on the Diversity and Composition of Fecal Microbiota

This study used 16S rRNA gene sequences from 60 fecal samples of Duroc pigs collected across three time-points (180 records) of the growing-finish period to determine the impact of prolonged stress challenge on the diversity and composition of pig fecal microbiota. After quality control, a total of 9.8 × 10^6^ reads were retained to detect 1124 Amplicon Sequence Variants (ASVs). In agreement with previous reports [25,40], the dominant bacterial phyla across the three time-points were Firmicutes and Bacteroidetes, and the five most abundant genera were *Lactobacillus*, *Limosilactobacillus*, *Clostridium*, *Streptococcus*, and *Prevotella* (Appendix A). Of the 1124 ASVs, 91.28% (1026 ASVs) were consistently identified across the three sampling points (Figure 3A). Sample distribution based on the Bray-Curtis distance matrix is represented in Figure 3B. Non-significant variations in Shannon-alpha diversity levels were observed between the experimental conditions (Figure 3C). Appendix A shows the observed patterns of alpha-diversity using Simpson, Chao1, and ACE indices. Beta-diversity did not show statistically significant differences between the groups (*p* = 0.06); however, there was a noticeable trend suggesting a more uniform fecal microbiota in the control group compared to the stress group at the end of the experiment (Figure 3D). 

The results of the PERMANOVA analysis indicate that under our experimental conditions, sex plays a minor role (*p* > 0.05) in shaping the structure of the microbial community. In addition, no significant differences in the abundance of genera were observed between females and castrated males. The number of bacterial biomarkers varied according to the sampling day: six at the first time-point (07_2022), 23 at the second (09_2022), and 15 at the last (10_2022). A detailed description of the bacterial biomarkers at each sampling time-point can be found in Appendix A. The topological parameters of inferred networks also differed between the two experimental conditions, with lower connectivity (node degree in control = 15.23 vs. stress = 13.81, *p* = 0.03) in the co-occurrence network derived from the stressed animal in comparison with the one from the control group (Appendix A).

In addition, the NetMoss2 pipeline allowed us to consolidate the genera abundance matrix generated across the experiment into two comprehensive, integrated co-occurrence networks (one for the control and one for the stress group), thus offering a more comprehensive picture of the microbial abundance dynamics throughout the growing-finish period. As illustrated in Figure 4, important distinctions between the stress and control groups were found, with 18 genera showing significantly differential abundance patterns and a moderate ability to distinguish between the two experimental conditions (AUC = 0.77). Compared to their control counterparts, the fecal microbiota of the stressed pigs exhibited a lower relative abundance of *Coprococcus* (FDR = 0.042), *DSXL01* (FDR = 0.021), *SFDB01* (FDR = 0.021), and *HUN007* (FRD = 0.023) genera. A contrasting trend was observed for the other 14 genera including the enrichment of the opportunistic bacteria *Holdemanella* (FDR = 0.034) and *Collinsella* (FDR = 0.097). The genera *Phascolarctobacterium* (FDR = 0.021), *Megasphaera* (FDR = 0.015), *Intestinibacter* (FDR = 0.021), and four members of family *Lachnospiraceae* [*Blautia* (FDR = 0.063), *Bariatricus* (FDR = 0.015), *Dorea* (FDR = 0.022), and *Marvinbryantia* (FDR = 0.049)] were also enriched in the stress group. 

## 4. Discussion

The results of this study offer a comprehensive understanding of the relationship between social stress in pigs and its effects on animal performance, behavior, and microbiota during the growing-finish period. 

In the present study, two sources of stress were combined; mixings of unknown animals and reduced space allowance. Previous research has shown that the regrouping of pigs typically results in intense fighting as they establish a new dominance hierarchy, causing significant stress to the animals [41]. This often results in wounds and depressive-like performance including elevated plasma cortisol concentrations, affecting immune function [42], metabolic, and endocrine responses in pigs negatively [43]. In fact, mixed pigs have poorer growth performance [44]. The smaller the floor space, the more aggressions occur after mixing animals, which affects the formation of a stable hierarchy [45]. Mixtures also present a long-term effect, in addition to the already known acute stress effect [44]. Space allowance affects competition for resources by limiting access to food, water, and resting areas, among other factors [6,45], especially when animals become older due to their higher body weight, which leads to a decrease in ADG and increase in FCR (9). Moreover, this competition for space is an excellent long-term stressor, since it provides an increment of aggressions and injuries [46,47]. Therefore, the pig model used in the present study, already used in the past by Fonseca et al. [29,48], combined acute and chronic stress factors. In addition, consistent with previous reports, our results confirm that prolonged social stress during the growing period significantly influences pig productivity, including a reduction in body weight, average daily gain, and feed efficiency, impacting feeding behavior and altering the fecal microbial ecosystem [28,49,50,51]. In any case, it is important to note that other sources of stress could have other effects on the animals’ physiology and microbiota, so the results found in the present study should not be generalized to any stress factor affecting pigs.

Moreover, in agreement with previous reports [16,19,28,52] no stress impact on the fecal microbiota’s Shannon-alpha diversity was perceived (Figure 3). Notably, 91.28% of the total ASVs were observed to overlap across the three sampling points (Figure 3A). This fact may suggest that the stress challenge prompted differences in the relative abundance of species rather than in species richness between the experimental conditions. Several bacterial biomarkers were identified, with a higher number detected at the second sampling time (09_2022), as compared to both the initial (07_2022) and final (10_2022) samplings (Appendix A). We hypothesize that observed differences between time-points could be attributed to the proximity between the mixed-stress event and sampling collection (17 days after the mix for the second sampling, as opposed to 51 and 50 days for the first and last samplings, respectively). In fact, this second sampling was selected to assess this acute effect of the social stress model, while the third (one month later) was selected to assess a more chronic effect. In any case, similar results have been reported in mice where stress significantly modulates the microbiota, mainly when the samples were assessed closely after stressor exposure [53]. We also observed that 56.52% of the genera identified at the second time-point were not detected in the integrated analysis or any of the two additional sampling points. This suggests potential changes in the gut microbiota, particularly involving stress-enriched bacteria. These include opportunistic genera like *Bulleidia*, *Streptococcus*, *Treponema*, and *UBA636* (a member of the *Erysipelotrichaceae* family). Conversely, a reduction in the abundance of beneficial bacteria such as *Anaerobutyricum*, *Enterenecus*, or *Peptococcus* was observed. These findings may be indicative of fecal microbial signatures associated with acute or short-term stressful situations. Consistent with our findings, a recent study reported the enrichment of opportunistic genera *Streptococcus* and *Treponema* in response to a 28-day social stress in growing pigs [28]. The longitudinal dynamics of the abundance of the bacteria mentioned above are shown in Appendix A. After the shifts caused by the acute stress episode, the relative abundance of these genera tends to recover close to its original values at the third sampling point, demonstrating the resilience of the microbiota after a perturbation. On the other hand, as three different diets were used during this study (Appendix A), some effect of this on the differences observed between the sampling periods cannot be ruled out.

Microbial signatures characterized by a reduction in the abundance of *Coprococcus*, *DSXL01*, *SFDB01*, and *HUN007* (a member of family Ruminococcaceae), along with the enrichment of *Collinsella*, *Phascolarctobacterium*, *Marvinbryantia*, and *ER4* genera were consistently observed in both the integrated analysis and at least two of the three sampling points, each separated by a minimum of 42 days. Hence, we propose these bacterial signatures as fecal indicators after prolonged stressful stimuli during the growing-finish period in pigs. As previously documented and confirmed by our findings, stress challenges prompt changes in the microbiota composition. Gut inflammation caused by stress may lead to the proliferation of pathogenic or opportunistic bacteria [15]. This could also be explained by the higher abundance of *Bulleidia*, *Streptococcus*, *Treponema*, and members of the family *Erysipelotrichaceae* in the group of stressed pigs. As previously reported by Nguyen et al., *Streptococcus* and *Treponema* were enriched in their stressed porcine model. The increase of *Streptococcus* abundance has been previously associated with major depressive disorder (MDD), and could play a role in the modulation of inflammatory response in depression patients [54,55]. Regarding feed efficiency, Kubasova et al. [56] studied the association between gut microbiota and RFI, showing that pigs with a high RFI (less efficient) had an increased abundance of *Erysipelotrichaceae* and *Collinsella*, and an increased abundance of *Streptococcus* after sanitary stress. Moreover, lower levels of members of *Erysipelotrichaceae* and *Streptococcus* were also reported in pigs with low RFI by McCormack et al. [57]. These results follow our findings; these three bacteria were enriched in the stress group, showing lower feed efficiency than the control group. Among the genera associated with long-term stress highlighted by the integrated analysis, we focus on the genera *Collinsella*, *Holdemanella*, *Phascolarctobacterium*, *Marvinbryantia*, and *ER4*. *Collinsella*’s pathobiont nature is related to several human diseases, like irritable bowel syndrome [58], Alzheimer’s disease, and autism spectrum disorder [59]. The pathogenic potential of *Collinsella* may be attributed to its capacity to enhance gut permeability and inflammation [58,60]. *Holdemanella*, a genus from the *Erysipelotrichaceae* family, was previously associated with a psychological stress model in rats [61], has recently been proposed as a microbial indicator of subordinance in pigs [62], and is linked to neurological disorders such as Alzheimer’s disease [63], and obsessive-compulsive behavior [64]. Even though the genus *Phascolarctobacterium* and members of the *Lachnospiraceae* family (*Blautia* and *Dorea*) were enriched in our stress group, previous studies have reported a negative association of these bacteria with psychological stress and potential beneficial effects for the host [65,66,67]. However, it is important to mention that observed patterns may be related to the individual’s resilience to stress. In mice, He et al. [68] revealed a correlation between stress resilience and mice microbiota composition in which stress-resistant mice exhibited lower levels of members of the family *Lachnospiraceae* including *Blautia* and *Roseburia* genera. Therefore, further investigation into the role of these genera is necessary to better understand their function under stress conditions. In line with our findings, a recent study [28] also observed an increase in *Marvinbryantia* abundance in a porcine stress model, while *ER4* has also been reported in many depression diseases, with an increased or reduced abundance depending on the study [54]. 

*Coprococcus*, a butyrate- and propionate-producing bacterium, is particularly interesting in terms of well-being indicators. Several lines of evidence across humans, murine models, and pigs support the decrease or depletion of *Coprococcus* because of different stressful scenarios. In humans, *Coprococcus* has been associated with a higher quality of life and is depleted in depression [69]. This genus was found also depleted in rats exhibiting depressive behavior after fecal transplantation from depressed human donors [70], and confirmed by meta-analysis as negatively associated with major depression and anxiety disorder in humans [55,71]. It is noteworthy that *Coprococcus* was the only genus consistently identified in our study as most abundant in the control group compared with the stress group throughout the experiment (Appendix A). This result is in line with a recent report showing the reduction of *Coprococcus* levels in the colon of crossbreed stressed male pigs [28]. *HUN007*, another genus linked to well-being, produces succinate [72,73], and belongs to the family Ruminococcaceae. Previous evidence supports the depletion of *HUN007* in MDD [19] and other neurological disorders like schizophrenia [72]. 

Previously mentioned microbial signatures, which we propose as indicators of stress or well-being, could play an important role in livestock health, production, and even be applied to human research [74]. Observed shifts of fecal microbiota give us an approach to how the microbial ecosystem evolves under stress conditions. Likewise, acquired knowledge of enriched stress bacteria could be relevant for monitoring animal welfare and the detection of stressed pigs on farms. Our approach allows the identification of non-invasive fecal biomarkers that after standardization could provide valuable information regarding the welfare status of the animals. For example, this can be achieved by quantitative real-time PCR or target sequencing approaches to quantify their prevalence, allowing rapid and accurate monitoring of animal welfare. Moreover, our results could be a starting point for proposing a microbial consortium that could be employed to reduce the negative effect of stressful stimuli and improve mood and productivity during the porcine growing-finish period. However, it is important to note that the model used in the present study addresses a very specific type of social stress that combines two different management strategies, a reduced space allocation that will impact animals chronically by increasing competition for space and mixing animals twice during the growing period that affects animal welfare more acutely. Although it is impossible to separate both factors, because the stress treatment is based on the combination of both, the result on the microbiota shows how some effects are maintained over time (so they could be related to the chronic consequences of the model) and others disappear after the mixtures (so they could be related to the acute consequences of the model). Furthermore, Ochoteco et al. [62] found how another intrinsic factor of social stress (in this case not controlled by the researchers) such as being subordinate or dominant within the group also impacts the microbiota, characterized by a higher abundance of beneficial genera *Faecalibacterium* and *Peptococcus* and a lower prevalence of *Holdemanella* in the microbial ecosystems of dominant pigs compared with subordinates. Therefore, it is evident that social stress caused by mixing, competition for space, or even social status, in addition to affecting welfare and performance in pigs, produces changes in the microbiota, mainly focused on an increase in opportunistic microorganisms and reduction in SCFA producers in the social stressed animals. However, although we observed that the gut microbial ecosystem of pigs exhibits resilience following disturbances caused by acute stress episodes such as mixing, our findings suggest potential fecal microbial biomarkers that may be associated with long-term stress conditions, acute stress episodes, and overall well-being. Further validation is necessary to determine at what level these changes are applicable to other types of stressors and can be used as an assessment tool for global animal welfare statuses.

Future research will be focused on investigating the effects of this consortium in a targeted manner using various animal models, including germ-free mice and different breeds of pigs. Once this methodology has been validated with these models, we propose advancing to studies related to human health, using the pig as a model to better understand the microbiome-gut-brain axis, and to mitigate the adverse effects of neurological disorders and proposed potential psychobiotics. A plausible mechanism of action could be through the SCFA production capacity of most well-being indicators, including butyrate, propionate, acetate, and succinate. A recent report showed that a higher relative abundance of butyrate producers is associated with a lower risk of hospitalization for infections [75]. Moreover, consistent with our study, stress leads to a depletion of butyrate producers with a subsequent enrichment of opportunistic bacteria, which may be detrimental to animal health [28]. SCFA-producing bacteria have been previously reported to be involved in developing resilience to stress [76], and the administration of SCFA to stressed mice helped to reduce long-lasting changes in mood, sensitivity to stress, and increased leakiness in the gut caused by stress [77]. Another modulatory strategy could be through dietary interventions using prebiotics, fecal microbiota transplantation, or the administration of psychobiotics that can restore abnormalities in the gut microbiota and abnormal brain function through the brain-gut-microbiota axis [78]. Finally, our results suggest the benefits of employing the pig as a biomedical model to gain deeper insights into the impact of social stress on humans. The striking parallels between our findings and alterations in human gut microbiota during periods of prolonged stress or neurological disorders highlight the relevance of a porcine model to understand better the role of the gut-brain axis with a dual focus on animal welfare and human well-being.

## 5. Conclusions

Our findings underscore the impact of prolonged stress on various aspects of pig performance, feeding behavior, and the composition of porcine fecal microbiota. Phenotypically, the social stress assessed in the present study manifests in reductions in body weight, average daily gain, and feed efficiency. The fecal microbiota analysis of stressed pigs revealed an enrichment of potentially opportunistic bacteria, including *Streptococcus*, *Treponema*, and several members of the *Erysipelotrichaceae* family, alongside a reduction of beneficial SCFA-producing bacteria. The longitudinal evaluation throughout the growing-finish period enables us to propose non-invasive fecal microbial biomarkers reflecting short and long-term stress events. Monitoring shifts of these biomarkers could enhance the assessment of pig welfare. The identified bacterial indicators of welfare may serve to design microbial consortia to alleviate stress during the growing period, thereby improving pig performance, welfare, and health. Additionally, some of these biomarkers hold promise as potential psychobiotics, offering a model for studying gut-brain axis disorders in humans, therefore, in the context of One-health, opening the possibility to impact both veterinary and biomedical research for pigs and humans.

## Figures and Tables

**Figure 1 animals-14-02704-f001:**
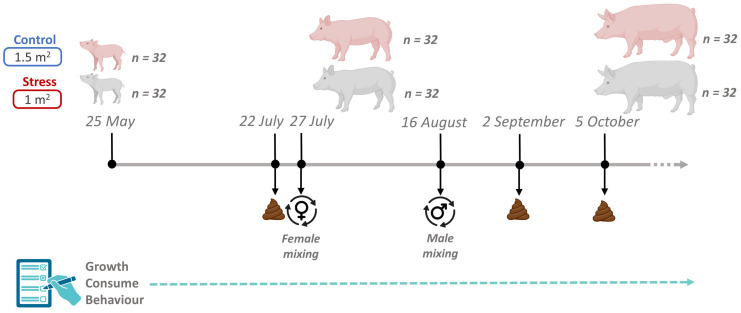
Overview of the experimental design, including sampling time-points, sample collection, and group mixing details.

**Figure 2 animals-14-02704-f002:**
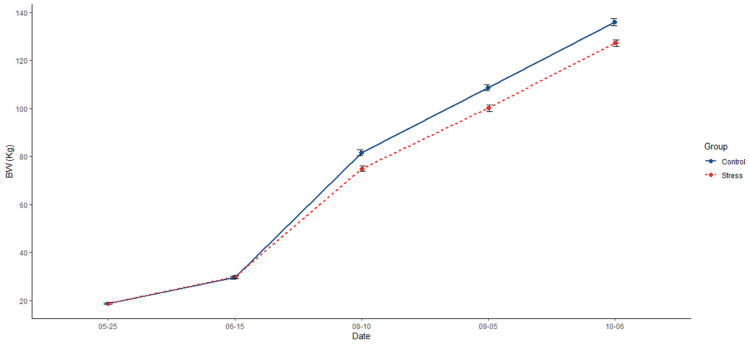
Comparative evolution across the experiment of mean of body weight (BW) and the standard error between control (solid blue line) and stress (dotted red line).

**Figure 3 animals-14-02704-f003:**
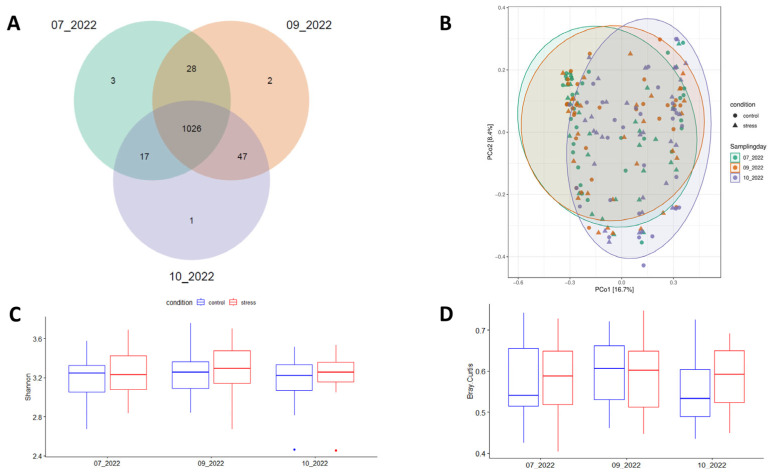
Number of ASVs commonly identified across the three sampling times (**A**), principal coordinates analysis (PCoA) plot based on the Bray-Curtis distance between samples (**B**), mean patterns of diversity index including Shannon-alpha (within samples) (**C**), and (**D**) Beta-diversity (between samples).

**Figure 4 animals-14-02704-f004:**
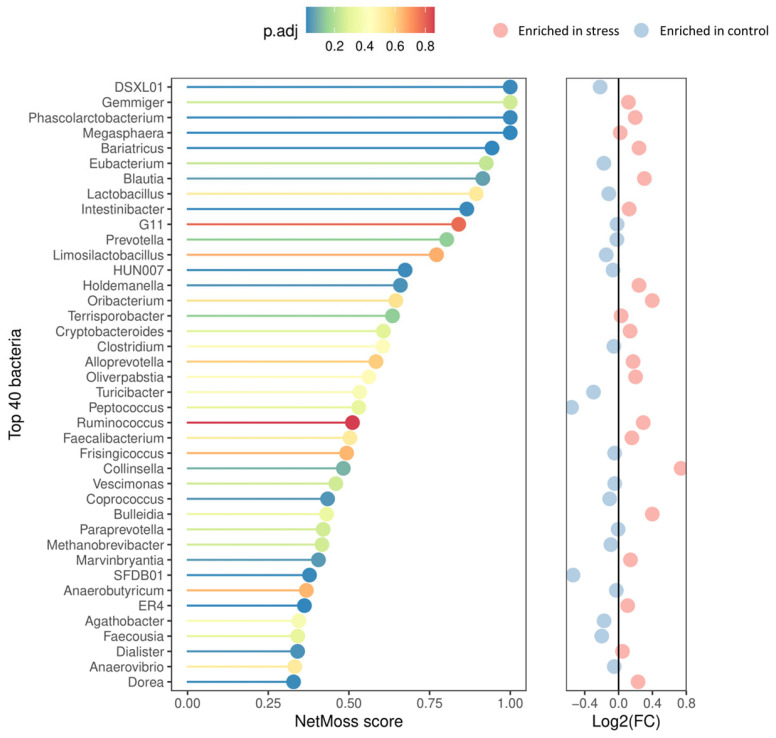
Top 40 genera according to NetMoss2 score and their corresponding abundance patterns. The corrected *p*-values are color-coded from blue (indicating the highest significance) to red (indicating no statistical significance). The adjacent panel on the right represents the differential abundance patterns, with red circles denoting genera enriched in the stress group and blue ones enriched in the control group.

**Table 1 animals-14-02704-t001:** Summary of mean phenotype differences and standard error (se) between control and stress group along the experiment.

Trait	Control (se)	Stress (se)	*p*-Value
Body weight (**BW** kg)	136 (1.49)	127 (1.35)	<0.0001
Average daily gain (**ADG** kg)	1.01 (0.01)	0.95 (0.01)	0.0001
Feed conversion ratio (**FCR** kg)	2.93 (0.18)	2.87 (0.16)	0.048
Total number of visits	719 (0.75)	589 (0.54)	<2 × 10^−16^
Total feed time per day (min)	74.55 (3.61)	75.83 (3.97)	0.028

## Data Availability

The entire data sets have been submitted to the NCBI’s sequence read archive with the BioProject accession number: PRJNA1131288.

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
