# Peer review of "Effect of Group Mixing and Available Space on Performance, Feeding Behavior, and Fecal Microbiota Composition during the Growth Period of Pigs"

_animals, 2024, doi:10.3390/ani14182704_

Round 1
Reviewer 1 Report
Comments and Suggestions for Authors
Points for improvement or doubts:
- Make clear the size of the pens, number of males and females per pen, and the number of pens for each treatment.
- Illustrate with images or figures how the animals were mixed in the stress treatment.
- Throughout the study, three diets were used that contained representative changes in the inclusion of macroingredients. This can alter the substrate profile for microbial development and cannot be neglected. Based on this understanding, the question arises: would there be a confounding effect on the changes in diversity and composition of the fecal microbiota considering stress and diet change? This needs to be discussed.
Author Response
Reviewer 1
Points for improvement or doubts:
Reviewer: - Make clear the size of the pens, number of males and females per pen, and the number of pens for each treatment.
Authors: All this part of material and methods was reworded to clarify these key points and now is explained as its follow (Lines 125-141): The experiment lasted 131 days, from June (animals were 60 days old) to October. The average weight of the animals at the arrival was 18 kg, being 18.8 ± 1,78 kg for males and 18.6 ± 1.64 kg for females, and they were maintained until the 140 kg. The 64 pigs were housed in the same building, in a total of 8 pens (4 control and 4 stress) containing 8 pigs each (4 castrated males and 4 females). The control pens had a total space of 12 m2 (1.5 m2 per pig) and the stress pens a total space of 8 m2 (1 m2 per pig). In both cases they had the same type of electronic feeding station and a drinker. In addition, all pens had two pieces of wood as enrichment material. In the four pens considered as stress treatment, animals were mixed twice during the growing period. The first mixing of animals for the stress treatment, which consisted of moving just the females among pens, was performed 61 days after the start of the study, and the second one, which consisted of moving just the males among pens, was performed 22 days later. In both cases, for males and females, three animals were changed to the other three pens (one per pen), and the fourth, the smallest female and the smallest male, remained in the original pen. In this way, mixed pigs did not meet again with their original pen mates (Figure 1). All animals had ad libitum access to a commercial cereal-based diet consisting in a starting, growing and finishing formula (Supplementary table 1).
Reviewer: Illustrate with images or figures how the animals were mixed in the stress treatment.
Authors: Thank you for the suggestion, we have create a new figure including how the animals were mixed.
Reviewer: Throughout the study, three diets were used that contained representative changes in the inclusion of macroingredients. This can alter the substrate profile for microbial development and cannot be neglected. Based on this understanding, the question arises: would there be a confounding effect on the changes in diversity and composition of the fecal microbiota considering stress and diet change? This needs to be discussed.
Authors: Thank you for the comment. In fact, the changes on the diet were done at the same time and in the same way in control and stressed pigs, so for this specific factor we can not expect any effect. However, it is true that within each treatment and along the time, any effect in the microbiota could be affected for the fact that the three diets differed along the study and we inclused a sentence about this factor in the discussion (Lines 370-372). On the other hand, we used the pigs and the diets provided by an important pig producer in Spain and we followed the commercial strategy in terms of nutrition that optimise the health status of the animals and this include three diets along the growing period. To maintain the same diet during the whole study would be against the nutritional logic of these animals, as the needs at the begining when they are young or at the finishing period, after reaching the 100 kg are not comparable, but it is true that now we need to consider this confounding factor in the discussion.
Reviewer 2 Report
Comments and Suggestions for Authors
This is a very good study, systematically researched on effect of mixing and available space on performance, feeding behaviour and fecal microbiota composition during the growth period of pigs, this is very meaningful for production in pigs. But some modifications are still needed to improve the quality of the paper.
1. The introduction section lacks a comprehensive description of the progress and does not provide an important explanation for the innovation of this paper.
2. The experimental details in the materials and methods section are not detailed enough, and the author needs to supplement more experimental details to enhance the credibility of the paper.
3. The discussion section of the paper did not provide a more extensive and in-depth discussion. Please ask the author to supplement it.
In summary, this paper is an excellent work and may be considered for publication after the author's revisions.
Author Response
Reviewer 2
This is a very good study, systematically researched on effect of mixing and available space on performance, feeding behaviour and fecal microbiota composition during the growth period of pigs, this is very meaningful for production in pigs. But some modifications are still needed to improve the quality of the paper.
Reviewer: The introduction section lacks a comprehensive description of the progress and does not provide an important explanation for the innovation of this paper.
Authors: We agree that it was not enough clear which was the innovative approach of this paper, and for this reason we reworded the end of the introduction to: “In the present study, it is used a social stress model in pigs based in two handling procedures in half of the pigs tested, to mix them twice during the growing period and to maintain a reduced space allowance to induce more competition for resources in comparison to the other half, with more space available and not mixed. This stress model was already tested with success in a previous study of our group , where it was found that growing rates differ dramatically between stressed and not stressed pigs. Therefore, although behavioural and phenotypic differences are also expected in the present study, its main objective is to check the fecal microbiota impacts of this long-term stress model in pigs to identify potential non-invasive fecal microbial biomarkers indicative of animal welfare. These biomarkers could offer insights into the dynamics of the gut microbial ecosystem under prolonged stress situations, to better understand the gut-brain axis, potentially leading to applications in both livestock production and even human health.
Reviewer: The experimental details in the materials and methods section are not detailed enough, and the author needs to supplement more experimental details to enhance the credibility of the paper.
Authors: We have reworded the first part of material and methods to solve this problem (Lines 125 - 141): In the present study, it is used a social stress model in pigs based in two handling procedures in half of the pigs tested, to mix them twice during the growing period and to maintain a reduced space allowance to induce more competition for resources in comparison to the other half, with more space available and not mixed. This stress model was already tested with success in a previous study (48) , where it was found that growing rates differ dramatically between stressed and not stressed pigs. Therefore, although behavioural and phenotypic differences are also expected in the present study, its main objective is to check the fecal microbiota impacts of this long-term stress model in pigs to identify potential non-invasive fecal microbial biomarkers indicative of animal welfare. These biomarkers could offer insights into the dynamics of the gut microbial ecosystem under prolonged stress situations, to better understand the gut-brain axis, potentially leading to applications in both livestock production and even human health.
Reviewer: The discussion section of the paper did not provide a more extensive and in-depth discussion.
Authors: We included a new section in the discusion (434 - 455): However, it is important to note that the model used in the present study addresses a very specific type of social stress that combines two different management strategies, a reduced space allocation that will impact animals chronically by increasing competition for space and mixing animals twice during the growing period that affects animal welfare more acutely. Although it is impossible to separate both factors, because stress treatment is based in the combination of both, the results on the microbiota shows how some effects are maintained over time (so they could be related to the chronic consequences of the model) and others disappear after the mixtures (so they should be related to the acute consequence of the model). Furthermore, Ochoteco et al. (63) found how another intrinsic factor of social stress (in this case not controlled by the researchers) such as being subordinate or dominant within the group is also impacting the microbiota in this case characterized by a higher abundance of beneficial genera Faecalibacterium and Peptococcus and a lower prevalence of Holdemanella in the microbial ecosystems of dominant pigs compared with subordinates. Therefore, it is evident that social stress, caused by mixing, competition for space or even the social status, in addition to affecting welfare and performance in pigs, produce changes in the microbiota. However, although we observed that the gut microbial ecosystem of pigs exhibits resilience following disturbances caused by acute stress episodes, such as mixing, our findings suggest potential fecal microbial biomarkers that may be associated with long-term stress conditions, acute stress episodes, and overall well-being, but further validation is necessary to determine at what level these changes are applicable to other type of stressors and to be used as an assessment tool for global animal welfare statuses.
Reviewer 3 Report
Comments and Suggestions for Authors
Dear Authors,
Your study provides valuable insights into how mixing and available space influence pig performance, feeding behavior, and fecal microbiota composition during the growth period. Your findings are well-presented and offer practical implications for welfare considerations for fattening pigs. However, further details on the specific interactions between mixing strategies and space availability, as well as their combined effects on microbiota, would enhance the understanding of their impact on overall pig health. I have one question and one suggestion regarding the study. That being said, I would recommend the authors specify the particular stress-related factors or changes in the gut microbiota that were identified as most significant in impacting the fattening phase of the pigs. The abstract and conclusion, highlight the particular modifications in these parameters linked to various mixing techniques and spatial circumstances, as well as how these elements interact to affect the general health and productivity of pigs in the fattening phase.
Thank you.
Author Response
Reviewer 3
Reviewer: Your study provides valuable insights into how mixing and available space influence pig performance, feeding behavior, and fecal microbiota composition during the growth period. Your findings are well-presented and offer practical implications for welfare considerations for fattening pigs. However, further details on the specific interactions between mixing strategies and space availability, as well as their combined effects on microbiota, would enhance the understanding of their impact on overall pig health. I have one question and one suggestion regarding the study. That being said, I would recommend the authors specify the particular stress-related factors or changes in the gut microbiota that were identified as most significant in impacting the fattening phase of the pigs. The abstract and conclusion, highlight the particular modifications in these parameters linked to various mixing techniques and spatial circumstances, as well as how these elements interact to affect the general health and productivity of pigs in the fattening phase.
Authors: We added some text in the discussion to solve this (Lines 434 - 455): However, it is important to note that the model used in the present study addresses a very specific type of social stress that combines two different management strategies, a reduced space allocation that will impact animals chronically by increasing competition for space and mixing animals twice during the growing period that affects animal welfare more acutely. Although it is impossible to separate both factors, because stress treatment is based in the combination of both, the results on the microbiota shows how some effects are maintained over time (so they could be related to the chronic consequences of the model) and others disappear some time after the mixtures (so they should be related to the acute consequence of the model). Furthermore, Ochoteco et al. (63) found how another intrinsic factor of social stress (in this case not controlled by the researchers) such as being subordinate or dominant within the group is also impacting the microbiota characterized by a higher abundance of beneficial genera Faecalibacterium and Peptococcus and a lower prevalence of Holdemanella in the microbial ecosystems of dominant pigs compared with subordinates. Therefore, it is evident that social stress, caused by mixing, competition for space or even the social status, in addition to affecting welfare and performance in pigs, produce changes in the microbiota, mainly focused on an increase in opportunistic microorganisms and reduction in SCFA producers in the social stressed animals. However, although we observed that the gut microbial ecosystem of pigs exhibits resilience following disturbances caused by acute stress episodes, such as mixing, our findings suggest potential fecal microbial biomarkers that may be associated with long-term stress conditions, acute stress episodes, and overall well-being, but further validation is necessary to determine at what level these changes are applicable to other type of stressors and to be used as an assessment tool for global animal welfare statuses.
Reviewer 4 Report
Comments and Suggestions for Authors
The study investigates the effects of prolonged social stress on pigs, revealing significant impacts on animal performance, feeding behavior, and gut microbiota composition. Overall, the paper is a strong study with minor issues in presenting its results and claims. The most interesting finding is the indication that fecal microbial biomarkers may be associated with animal well-being.
I have the following suggestions for the authors:
-
Line 2: The term "mixing" in the title could be vague; I suggest using "group mixing" for clarity.
-
Lines 27 and 42-43: References to "human gut-brain axis research" should be followed by a brief explanation connecting it to the study's topic.
-
Lines 30-31: The sentence “...divided in two experimental groups belonging to stress (n=30) and control (n=30)” is unclear and grammatically incorrect. Rephrase to: “...divided into two experimental groups: a stress group (n=30) and a control group (n=30).”
-
Line 35: The phrase “resilience patterns” in the abstract is vague. Please clarify what is meant by resilience patterns.
-
Line 90: Correct the spacing before the commas in "birth ," and "weaning ,".
-
Line 121: The section lacks details on animal husbandry practices, such as environmental conditions, housing type, and health monitoring protocols. These should be briefly provided to aid replicability.
-
Lines 129 and 138: The terms "males" and "castrated males" create inconsistency. Use uniform wording for clarity.
-
Lines 129-136: The setup of mixing arrangements is unclear. Specify the number of pigs per pen and the number of pens.
-
Line 166: The fecal sampling time points (July 22nd, September 2nd, October 5th) may not adequately capture short-term stress dynamics due to the large gap between the second and third sampling. Provide a rationale for this sampling schedule in the discussion.
-
Lines 250-315: Address why dominant pigs in the stress group reacted similarly to less dominant pigs. Provide an explanation in the results or discussion.
-
Lines 278-285: Provide a more detailed explanation of Figure 3, particularly for panels A and B, and ensure this is referenced in the discussion.
-
Lines 283-284: A p-value of 0.06 should not be described as a "statistical trend." State explicitly that the result is not statistically significant: "The beta-diversity did not show statistically significant differences between the groups (p=0.06)."
-
Lines 325-326: The statement about proposing fecal microbial biomarkers for various stress conditions is an overstatement. Amend to: "Our findings suggest potential fecal microbial biomarkers that may be associated with long-term stress conditions, acute stress episodes, and overall well-being, but further validation is necessary."
Author Response
Reviewer 4
The study investigates the effects of prolonged social stress on pigs, revealing significant impacts on animal performance, feeding behavior, and gut microbiota composition. Overall, the paper is a strong study with minor issues in presenting its results and claims. The most interesting finding is the indication that fecal microbial biomarkers may be associated with animal well-being.
I have the following suggestions for the authors:
Reviewer: Line 2: The term "mixing" in the title could be vague; I suggest using "group mixing" for clarity.
Authors: Thank you for the suggestion, we have modified it.
Reviewer: Lines 27 and 42-43: References to "human gut-brain axis research" should be followed by a brief explanation connecting it to the study's topic.
Authors: Thank you for the suggestion. We have modified the corresponding sentences(Lines 26, and 39 - 41)
Reviewer: Lines 30-31: The sentence “...divided in two experimental groups belonging to stress (n=30) and control (n=30)” is unclear and grammatically incorrect. Rephrase to: “...divided into two experimental groups: a stress group (n=30) and a control group (n=30).”
Authors: All this part of the text was reworded (Lines 29 -30).
Reviewer: Line 35: The phrase “resilience patterns” in the abstract is vague. Please clarify what is meant by resilience patterns.
Authors: Here, we refer to the capacity of the gut microbiota to return to its stable state after a perturbation (i.e. stressful events) commonly known as microbiota resilience. We have modifed the sentence, but due to limitations of the number of words we cannnot define resilicence here.
Reviewer: Line 90: Correct the spacing before the commas in "birth ," and "weaning ,".
Authors: Done
Reviewer: Line 121: The section lacks details on animal husbandry practices, such as environmental conditions, housing type, and health monitoring protocols. These should be briefly provided to aid replicability.
Authors: This part of the material and methods was reworded accordingly (Lines 125 -141).
Reviewer: Lines 129 and 138: The terms "males" and "castrated males" create inconsistency. Use uniform wording for clarity.
Authors: Thank you for the suggestion, we modified it for consistency to castrated males.
Reviewer: Lines 129-136: The setup of mixing arrangements is unclear. Specify the number of pigs per pen and the number of pens.
Authors: All this part has been reworded and now includes: In the present study, it is used a social stress model in pigs based in two handling procedures in half of the pigs tested, to mix them twice during the growing period and to maintain a reduced space allowance to induce more competition for resources in comparison to the other half, with more space available and not mixed. This stress model was already tested with success in a previous study , where it was found that growing rates differ dramatically between stressed and not stressed pigs. Therefore, although behavioural and phenotypic differences are also expected in the present study, its main objective is to check the fecal microbiota impacts of this long-term stress model in pigs to identify potential non-invasive fecal microbial biomarkers indicative of animal welfare. These biomarkers could offer insights into the dynamics of the gut microbial ecosystem under prolonged stress situations, to better understand the gut-brain axis, potentially leading to applications in both livestock production and even human health.
Reviewer: Line 166: The fecal sampling time points (July 22nd, September 2nd, October 5th) may not adequately capture short-term stress dynamics due to the large gap between the second and third sampling. Provide a rationale for this sampling schedule in the discussion.
Authors: We added a new sentence in the discussion addressing this point (line 354-356).
Reviewer: Lines 250-315: Address why dominant pigs in the stress group reacted similarly to less dominant pigs. Provide an explanation in the results or discussion.
Authors: We added this new text in the discussion that include this point (Lines 434 - 455): However, it is important to note that the model used in the present study addresses a very specific type of social stress that combines two different management strategies, a reduced space allocation that will impact animals chronically by increasing competition for space and mixing animals twice during the growing period that affects animal welfare more acutely. Although it is impossible to separate both factors, because stress treatment is based in the combination of both, the results on the microbiota shows how some effects are maintained over time (so they could be related to the chronic consequences of the model) and others disappear some time after the mixtures (so they should be related to the acute consequence of the model). Furthermore, Ochoteco et al. (63) found how another intrinsic factor of social stress (in this case not controlled by the researchers) such as being subordinate or dominant within the group is also impacting the microbiota in this case characterized by a higher abundance of beneficial genera Faecalibacterium and Peptococcus and a lower prevalence of Holdemanella in the microbial ecosystems of dominant pigs compared with subordinates. Therefore, it is evident that social stress, caused by mixing, competition for space or even the social status, in addition to affecting welfare and performance in pigs, produce changes in the microbiota, mainly focused on an increase in opportunistic microorganisms and reduction in SCFA producers in the social stressed animals. However, although we observed that the gut microbial ecosystem of pigs exhibits resilience following disturbances caused by acute stress episodes, such as mixing, our findings suggest potential fecal microbial biomarkers that may be associated with long-term stress conditions, acute stress episodes, and overall well-being, but further validation is necessary to determine at what level these changes are applicable to other type of stressors and to be used as an assessment tool for global animal welfare statuses.
Reviewer: Lines 278-285: Provide a more detailed explanation of Figure 3, particularly for panels A and B, and ensure this is referenced in the discussion.
Authors: We have include a sentence in the discussion (Lines 346 -347)
Reviewer: Lines 283-284: A p-value of 0.06 should not be described as a "statistical trend." State explicitly that the result is not statistically significant: "The beta-diversity did not show statistically significant differences between the groups (p=0.06)."
Authors: Thank you for your comment. We acknowledge that a p-value of 0.06 does not meet the conventional threshold for statistical significance. However, we believe it is still important to report the observed pattern in beta-diversity. We have revised the statement accordingly to clarify the lack of statistical significance while still noting the observed trend (Lines 281 -284)
Reviewer: Lines 325-326: The statement about proposing fecal microbial biomarkers for various stress conditions is an overstatement. Amend to: "Our findings suggest potential fecal microbial biomarkers that may be associated with long-term stress conditions, acute stress episodes, and overall well-being, but further validation is necessary."
Authors: We added this sentence in a new section of the discussion (Lines 434-455): However, it is important to note that the model used in the present study addresses a very specific type of social stress that combines two different management strategies, a reduced space allocation that will impact animals chronically by increasing competition for space and mixing animals twice during the growing period that affects animal welfare more acutely. Although it is impossible to separate both factors, because stress treatment is based in the combination of both, the results on the microbiota shows how some effects are maintained over time (so they could be related to the chronic consequences of the model) and others disappear some time after the mixtures (so they should be related to the acute consequence of the model). Furthermore, Ochoteco et al. (63) found how another intrinsic factor of social stress (in this case not controlled by the researchers) such as being subordinate or dominant within the group is also impacting the microbiota in this case characterized by a higher abundance of beneficial genera Faecalibacterium and Peptococcus and a lower prevalence of Holdemanella in the microbial ecosystems of dominant pigs compared with subordinates. Therefore, it is evident that social stress, caused by mixing, competition for space or even the social status, in addition to affecting welfare and performance in pigs, produce changes in the microbiota, mainly focused on an increase in opportunistic microorganisms and reduction in SCFA producers in the social stressed animals. However, although we observed that the gut microbial ecosystem of pigs exhibits resilience following disturbances caused by acute stress episodes, such as mixing, our findings suggest potential fecal microbial biomarkers that may be associated with long-term stress conditions, acute stress episodes, and overall well-being, but further validation is necessary to determine at what level these changes are applicable to other type of stressors and to be used as an assessment tool for global animal welfare statuses.